# Effects of an Adaptogenic Extract on Electrical Activity of the Brain in Elderly Subjects with Mild Cognitive Impairment: A Randomized, Double-Blind, Placebo-Controlled, Two-Armed Cross-Over Study

**DOI:** 10.3390/ph13030045

**Published:** 2020-03-14

**Authors:** Wilfried Dimpfel, Leonie Schombert, Ingrid K. Keplinger-Dimpfel, Alexander Panossian

**Affiliations:** 1Justus-Liebig-University Giessen, Giessen, c/o NeuroCode AG, D-35578 Wetzlar, Germany; w.dimpfel@neurocode-ag.com; 2NeuroCode AG, D-35578 Wetzlar, Germany; leonie@neurocode-ag.com (L.S.); keplinger@neurocode-ag.com (I.K.K.-D.); 3Phytomed AB, Bofinkvagen 1, 31275 Vaxtorp, Sweden

**Keywords:** elderly subjects, cognition, *Andrographis paniculata*, *Withania somnifera*, adaptogens, quantitative EEG, sleep, psychometry, discriminant analysis

## Abstract

Background: The current and potential uses of adaptogens are mainly related to treatment of stress-induced fatigue, impaired cognitive function, mental illness, and behavioral- and age-related disorders. However, clinical evidence regarding the efficacy of adaptogens is limited. The primary aim of this study is to determine whether a combination of adaptogenic plant extracts from *Andrographis paniculata* and *Withania somnifera* (Adaptra^®^ Forte) could be used as effective and safe treatment for impaired cognitive, memory, or learning ability functions and sleep disorders. Methods: The changes in electroencephalogram (EEG) frequency ranges in 17 different brain regions, psychometric tests of cognitive performance, as well as standard questionnaires of assessment of mood and sleep were measured after single and repeated administration of Adaptra^®^ or placebo for four weeks and after a two-week treatment-free follow-up period within a randomized, double-blind, placebo-controlled two-armed cross-over study. Results: Adaptra^®^ Forte significantly improved cognitive performance in the d2-Test for attention and the concentration performance test after four weeks’ treatment, and was positively correlated with increases in δ and θ power in the quantitative EEG compared with placebo during cognitive challenges. Conclusion: The results of this study suggest that Adaptra^®^ Forte exhibits a calming and anxiolytic effect without sedation, and is associated with overall stress-protective activity.

## 1. Introduction

Adaptogens are natural compounds or plant extracts that increase the adaptability, resilience, and survival of living organisms subject to stress [1,2,3]. During the last decade, many molecular targets, networks, and signaling pathways that are regulated by adaptogens have been identified [1,4,5,6,7,8,9,10]. The current and potential uses of adaptogens include treatment of stress-induced fatigue, impaired cognitive function, mental illness, and behavioral- and aging-related disorders [11,12,13,14,15,16,17,18,19,20,21,22,23,24,25,26,27,28,29,30,31,32,33].

Of particular interest are two adaptogenic plants, *Andrographis paniculata* and *Withania somnifera*, that have been extensively studied. However, there is limited evidence regarding the clinical efficacy of adaptogens as applied in treatment of cognitive impairment. A number of preclinical studies indicate synergistic interactions of various combinations of adaptogenic plants, e.g., *A. paniculata* with *W. somnifera* (Adaptra^®^ Forte). Growing in Southeast Asia (mainly in China, Thailand, India, and Nepal), *A. paniculata* (Burm. F. Nees*), (*Andrographidis* Kraut: Ger.) is one of the mostly used medicinal plants in Asia for treatment of respiratory infections, such as the common cold, influenza with fever, sore throat, acute and chronic cough, sinusitis, bronchitis, and pharyngotonsillitis [33,34,35]. The functional claims of *A. paniculata* dietary supplements (for a consolidated list of Article 13 health claims of the European Food Safety Authority (EFSA), please refer to http://www.efsa.europa.eu/EFSA/efsa_locale-1178620753812_article13.htm) are related mainly to the categories “respiratory health”, “immune function”, and “body defenses against external agents” [33,34,35]. Animal studies showed that andrographolide ameliorates the symptoms of experimental autoimmune encephalomyelitis [36]. Furthermore, treatment with *Andrographis* downregulates expression of *KIT*, *IRF1*, *CD24*, *CASP1*, *VCAM1*, *NFRSF14*, and *EPHA4* genes and upregulates expression *NRG1* and *NGF* genes, suggesting predated inhibition of encephalitis [37]. The main active ingredient, andrographolide, improved mediators of synaptic plasticity in the hippocampus and cerebral cortex and prevented neurodegeneration, suggesting potential use of andrographolide and *Andrographis* as neuroprotective agents for the management of neurological disorders associated with memory impairment [38].

Radix Withaniae consisting of the dried roots of *Withania somnifera* (L.) Dunal. (Solanaceae) is widespread from the Mediterranean coast to India in semi-arid habitats. Radix Withaniae, known as ‘Ashwagandha’ in Sanskrit and as ‘Indian ginseng’ in Ayurveda [25], is used both in Ayurveda and Unani traditional medicinal systems for the treatment of numerous diseases, including neurological and mental disorders. The anxiolytic- and antidepressant-like activities of the alcohol root extract and isolated glycowithanolides, sitoindosides, and withaferin A were demonstrated in animal models of depression and anxiety [26,27,28,29]. *Withania somnifera* root extract induced sedation, a reduction in locomotor activity, potentiation of thiopental-induced sleep, reduction of catecholamines and acetylcholine with an increase in histamine and serotonin in the brain tissue of mice, and a delay in semicarbazide- and pentylenetetrazole-induced seizures [29,30]. It is possible that GABAergic activity is the mechanism through which GABAergic signaling dysfunctions, such as in anxiety disorders and insomnia, are mitigated through treatment [31].

Seven clinical studies described *Withania somnifera* as a safe and effective adaptogen for treating stress-related neuropsychiatric conditions, such as anxiety and bipolar disorder [18]. For example, in a double-blind, randomized placebo-controlled trial (RPCT), 64 adult subjects exposed to chronic stress took one capsule twice daily of either *W. somnifera* extract (300 mg) or placebo for 60 days [32]. Telephone interviews were used to assess safety and compliance on days 15, 30, and 45, and at endpoint on day 60. At the study endpoint, subjects taking *W. somnifera* extract showed significantly greater reductions as measured on all stress scales versus placebo (*p* < 0.0001), and a significantly greater reduction in serum cortisol levels versus placebo (*p* = 0.0006). Adverse event rates were comparable in both groups.

The anxiolytic effects of five different preparations of *W. somnifera* were observed in five clinical studies using a total of 157 patients [18]. Three of these studies were double-blind RCPTs with quality scores of 2–3 out of 5. The limitations of these studies include insufficient details regarding preparation of the extract, treatment administration, and randomization procedures. The best study (quality score 4), was an eight-week double-blind RCT which used a water extract and 250–500 mg/day *Withania* (Sensoril^®^) in testing 60 subjects with bipolar disorder (DSM-IV). Compared with the control, subjects taking *W. somnifera* had significantly better performance on three cognitive tasks: digit span backward (*p* = 0.035), flanker neutral response time (*p* = 0.033), and social cognition response of the Penn Emotional Acuity Test (*p* = 0.045). Other cognitive tests were not significantly different between groups. Adverse events were similar in both groups. These encouraging results need to be confirmed in clinical trials of standardized products.

The present study was initiated to test a combination of *Andrographis paniculata* and *Withania somnifera* (the dietary supplement Adaptra^®^ Forte) in elderly subjects suffering from mild cognitive impairment. We hoped that the two preparations would interact positively with each other and provide adaptogenic efficacy within a clinical setting.

The primary aim of the study was to determine whether Adaptra^®^ Forte is an effective and safe treatment for impaired cognitive functions, memory, learning ability, sleep disorders, electrical activity of the brain (at six electroencephalogram (EEG) brain frequencies δ, θ, α1, α2, β1, and β2 in 17 different brain regions), and whether Adaptra^®^ Forte is superior to the placebo after single and repeated administration for four weeks. The secondary aim of the study was to assess the durability of the effect over a two-week-long, treatment-free follow-up period. This pilot study was designed to provide preliminary data to power a future large-scale study of Adaptra^®^ Forte efficacy for cognitive impairment.

## 2. Results

### 2.1. Baseline Data of Study Participants

Overall, 16 participants were assessed for eligibility; all met the inclusion criteria. Sixteen patients were recruited in the study, of which 15 patients completed treatment and were included in the final analysis. Baseline demographic and efficacy outcome measures are shown in Table 1.

### 2.2. Efficacy of Treatment

Efficacy of treatment was evaluated using three measures:Objective recording of quantitative EEG (efficacy primary outcomes) during performance of several psychometric tasks,Objective assessment of cognitive functions and mental performance in several psychometric tests (efficacy secondary outcomes), andSubjective assessment of mental health and quality of sleep based on validated questionnaires (efficacy secondary outcomes).

#### 2.2.1. Efficacy Primary Endpoint

The efficacy primary endpoint in this study was the change from baseline in neural electric activity of the brain when considering background mental activity after multiple doses of Adaptra^®^ Forte administration for four weeks compared with the placebo control.

The responses of electric brain activity were measured as spectral power in 17 different brain regions within six specially defined frequency ranges (δ, θ, α1, α2, β1, and β2) during relaxation as well as during tests of cognitive performance: test for attention (d2-Test), memory test (ME-Test), concentration performance test with financial reward (CPT).

##### Quantitative Electroencephalogram (EEG) Results after Acute and Repetitive Dosing

Electric brain activity was recorded under several different conditions. The first recording was always in a relaxed state with either open eyes or closed eyes. After this, three different mental challenges were presented during quantitative EEG recordings. EEG data were documented initially as absolute spectral power (µV^2^) for each electrode position (brain area) and each frequency range (δ–β2). There were only small differences between the three groups with respect to median values of spectral power in the six frequency ranges. Absolute power values from the baseline recording with respect to all recording conditions were then set as 100%. Drug-induced changes were documented as comparison of pre–post intake as a percentage of baseline values for each condition. An overview of the absolute spectral power values during recording in the relaxed state (eyes open and eyes closed) on the first day (acute) before taking the placebo or Adaptra^®^ Forte is given in Table 2 and Table 3. High values in frontal brain locations (F_7_ and F_8_) are normal and have been observed in every clinical study to date. When comparing the median values of both treatment groups, no major differences were recognized. Thus, the starting values are comparable.

To obtain a first overview on the effects of the placebo and Adaptra^®^ Forte on brain electricity, an overall effect was determined by averaging data from all electrode positions for each recording condition and each frequency band. During relaxation (recording condition eyes open), no statistically significant changes were recognized on the first experimental day at 2 h after intake in comparison to the placebo, despite the increase in δ waves. After four weeks of intake, a statistically significant (*p* < 0.05) increase of spectral θ power emerged 2 h after intake when recording with eyes open. Details are provided in Figure 1. During recording with eyes closed, a statistically conspicuous increase in θ power (*p* < 0.12) was recognized in comparison with the placebo on the first experimental day at 2 h after intake (Figure 1, acute). After four weeks, a highly significant increase in δ and θ power was documented at 2 h after intake on the last repetitive experimental day (*p* < 0.01). An increase of α1 power was statistically conspicuous (*p* < 0.12). Details are provided in Figure 1.

After setting spectral baseline values as 100%, drug-induced changes were documented for each electrode position and all frequencies at 2 h after intake of the trial drug. With respect to the placebo, we observed hardly any change during relaxation (eyes open recording condition), whereas with respect to Adaptra^®^ Forte, we observed statistically significant increases in spectral δ power in the parietal lobe on the first experimental day. In addition, at the parietal electrode position P_4_, we observed significant increases in spectral α and β power. In the temporal lobe, a conspicuous decrease of β1 power was noted, whereas in the occipital lobe, we found a significant decrease in β2 power (Figure 2). On the last experimental day after repetitive administration for four weeks, we again noted statistically significant δ power increases in central and parietal brain areas at only 2 h after intake of Adaptra^®^ Forte during relaxation. Statistically significant increases in spectral θ power were documented in five central, parietal, and temporal brain regions, as depicted in Figure 2. Statistically significantly increases in α1 spectral power only occurred at electrode position C_z_.

Effects of Adaptra^®^ Forte under the eyes closed recording condition during relaxation were more prominent. On the first experimental day, statistically significant increases in δ waves were seen in the parietal (electrode positions P_z_ and P_4_) and temporal area (electrode positions T_3_ and T_5_). Statistically significant increases in spectral θ power occurred in 8 out of 17 brain areas. Significant increases in α1 spectral power were only observed in the central area C_3_ and the occipital area O_1_. The observed changes in spectral power induced by Adaptra^®^ Forte were reproduced on the last experimental day. Statistically significant increases in spectral δ power appeared in 6 out of 17 brain areas. These increases emerged in central, parietal, and temporal brain regions. The same regions also displayed a statistically significant increase in spectral θ power. Significant changes in α1 power only occurred at electrode positions F_z_ and C_3_, as shown in Figure 3.

After conducting the EEG recordings in the relaxed state (eyes open and eyes closed), cognitive testing started. Three different psychometric tests were performed: d2-attention-Test, concentration performance test CPT (calculation test), and memory test. The spectral changes were evaluated in comparison with the relaxed eyes open state by setting those values obtained during relaxation (eyes open) as 100%. Thus, test-specific spectral changes were monitored. Looking at baseline data one day after repetitive intake for four weeks, the performance of the d2-attention-Test was accompanied by statistically conspicuous or significant increases in δ/θ spectral power in six brain regions in comparison to the placebo (Figure 4). Looking at baseline data one day after repetitive intake for four weeks, performance on the CPT was accompanied by statistically conspicuous or significant increases in δ/θ spectral power in five brain regions in comparison to the placebo (Figure 4). Looking at baseline data one day after repetitive intake for four weeks, the performance on the memory test only showed minor changes in spectral power with respect to δ/θ or α spectral power. Details are provided in Figure 4.

##### Follow-Up Two Weeks after the Last Experimental Day

To evaluate the spectral frequency pattern of the follow-up recording, data from day A were set as 100% and data from the follow-up recording were compared with the baseline of the first day. Differences in the first recording were noted for each brain region and each frequency for the eyes open recording condition and for the eyes closed recording condition (Figure 5). The data at the follow-up were comparable to the first experimental day and did not indicate any special effect induced by Adaptra^®^ Forte. 

With respect to cognitive challenges, test-specific spectral changes (eyes open condition set as 100%) did not show major differences between day A and the follow-up recording, even if single brain areas showed statistically significant deviations in comparison to day A. The overall pattern of spectral changes was similar. During the Concentration Performance test (CPT), somewhat lower δ and θ activities were recorded at the follow-up in comparison to day A (data not shown).

##### Efficacy of Adaptra^®^ Forte Documented by Discriminant Analysis

All 102 parameters from the EEG recording (17 electrode positions × 6 frequency ranges) during the eyes open condition were fed into a linear discriminant analysis containing data from a number of preparations tested earlier with identical methodology. The results of both experimental days were in close vicinity of each other and not well separated. They also showed a similar color (Adaptra Forte labeled as “ac” for acute and Adaptra Forte labeled “re” for repetitive in Figure 6). After repetitive intake, the data approached the projection place of the herbal drug Calmvalera^®^ (Hevert Arzneimittel, D 5559 Nussbaum, Germany), depicted in the upper part of the six-dimensional image. The stimulating preparations Zembrin^®^ (Brunel Laboratoria, Gauteng 0149, South Africa) and memoLoges^®^ (Dr. Loges + Co. GmbH, D 21,423 Winsen (Luhe), Germany) appears toward the front, whereas Adaptra^®^ Forte appears toward the back. Details are provided in Figure 6.

#### 2.2.2. Efficacy Primary Endpoints

##### Results from Performance of Psychometric Tests

In total, three psychometric tests were performed during quantitative EEG recording on the first and last visit: concentration (d2-attention-Test), memory test (ME-Test), and concentration performance test (CPT). No statistically significant differences were observed between the placebo or Adaptra^®^ Forte groups 2 h after intake in either the first visit or the last visit (data not shown). Comparison of the performance of the psychometric tests between the first and last experimental day revealed statistically significant differences between Adaptra^®^ Forte and placebo with respect to the d2-attention-Test. With respect to the memory test, the baseline value was significantly better in the Adaptra^®^ Forte group when the difference between the two days was evaluated. Details are provided in Table 4.

##### Subjective Assessment of Effect of Adaptra^®^ Forte Mood State and Quality of Sleep Based on Questionnaires

Two questionnaires were completed on the different study days: Profile of Mood States (POMS) and the sleep questionnaire SF-B/R (Schlaffragebogen). 

Four mood states were used for the evaluation of the POMS questionnaire: S1 = dejection, S2 = sullenness, S3 = fatigue, and S4 = thirst for action. Comparing the results obtained by the Profile of Mood States, we found no statistically significant differences with respect to the placebo and Adaptra^®^ Forte for the first or last experimental day (data not shown). Only a very slight increase in fatigue seemed to emerge on both days for Adaptra^®^ Forte in comparison with placebo (both baseline and placebo values were around 1, whereas values of 1.37 and 1.32 were recorded for Adaptra^®^ Forte, but these values were not statistically significant. Only the score “thirst for action” significantly increased after four weeks of intake compared to baseline (*p* = 0.05).

The following sub-scores were used for the evaluation of the sleep questionnaire SF-B/R: SQ = sleep quality; GES = feeling refreshed after sleep/feeling of being recovered after sleep; PSYA = psychic well-balanced feeling in the evening; PSYE = psychic exhausted feeling in the evening; PSS = psychosomatic symptoms during sleep; TRME = dream recall; and SWR = sleep–wake regulation. Details are provided in Table 5. Comparing the results of the sleep questionnaire, SF-B/R score values for sleep quality (SQ), refreshment (GES) sleep questionnaire and psychosomatic symptoms (PSS) were significantly different after four weeks’ intake of Adaptra^®^ Forte.

### 2.3. Safety Evaluation

Sixteen patients were administered Adaptra^®^ Forte, and fifteen patients were administered the placebo preparation at a daily dose of two capsules per day for four weeks. One participant dropped out during Phase B of the study.

In total, three adverse events (AEs) were observed in three patients during the treatment with Adaptra^®^ Forte: one allergic reaction in a patient using antihistamine cetirizine and hypertension in two subjects (12% of the sample size) concomitantly using hypotensive drugs “irbesartan” and “candesartan”. One of these patients discontinued the treatment after 32 days of repeated administration of Adaptra^®^ Forte. All adverse events presumably resulted from possible interactions with concomitant treatments. There were no serious adverse events or other significant side effects in this study. All subjects underwent a urine drug test (day of screening (SC)) as well as an alcohol test (day A, B, C, D, and follow-up (FU)) for safety. No deviations were registered during this clinical study. Vital signs (blood pressure and heart rate) were measured before each EEG recording (baseline 0 h) and 120 min (2 h) after intake of Adaptra^®^ Forte or Placebo. We recorded systolic blood pressure (SBP) and diastolic blood pressure (DBP) and pulse/heart rate (bpm), and physical examinations did not reveal any deviation from normality. No organic disease emerged during the study. All ECG measurements were within the normal range.

Overall, the treatment was well tolerated; no serious adverse events were detected in the presence of the dietary supplement Adaptra^®^ Forte capsules.

## 3. Discussion

We aimed to compare the effects of Adaptra^®^ Forte with a placebo using three different types of evidence: quantitative EEG recording in the presence of different challenges, psychometric testing, and questionnaires. Whereas no significant effects of Adaptra^®^ Forte were recognized with respect to the mood state using the POMS questionnaire, the sleep questionnaire showed statistically significant improvements (*p* < 0.05) with respect to sleep quality, refreshment, and occurrence of psychosomatic symptoms when the difference between the last experimental day was compared with the first experimental day. With respect to cognitive testing, our comparison of the difference between the first and last experimental days showed a statistically significant improvement (*p* < 0.05) according to the d2-attention-Test compared to baseline in the case of Adaptra^®^ Forte. Likewise, the difference in the baseline value between the first and last day was significantly better (*p* < 0.05) with respect to performance in the memory test for the Adaptra^®^ group. The results of cognitive testing could be confirmed in the future using a larger number of participants in a future study.

Quantitative EEG data revealed statistically significant effects in comparison with the placebo under different recording conditions. Typically, brain wave patterns reveal balances and imbalances within the brain, e.g., overactivity in certain brain sections may be correlated with the pharmacological effects of various drugs under various conditions [39].

During relaxation, increases in spectral δ and θ power emerged 2 h after intake of Adaptra^®^ Forte in numerous brain regions as early as the first experimental day. Spectral changes were more pronounced during the eyes closed recording condition than during the eyes open condition. After four weeks of repetitive intake of Adaptra^®^ Forte, spectral changes were even more pronounced in comparison with the first day.

With respect to spectral EEG changes during cognitive testing, only slight changes were observed 2 h after intake of Adaptra^®^ Forte (data not shown). Possibly due to the general increase in δ and θ spectral power during relaxation, an attenuation was observed for the Adaptra^®^ Forte in comparison to the placebo group, especially for the d2-attention-Test. Baseline values one day after continuous repetitive intake for four weeks showed a statistically significant focal increase in δ/θ power during performance of this test.

Drug-independent evidence from human EEG recordings [40] suggests that θ band activity increases when subjective alertness decreases in a given individual, even with moderate levels of sleepiness experienced during the daytime. Strong preclinical evidence for a calming action related to θ activity comes from the administration of drugs biochemically acting on norepinephrine α2 autoreceptors, like clonidine or medetomidine in rats [41]. This effect on θ spectral power was reproduced in humans using dexmedetomidine, which acts as an agonist on α2-adrenergic autoreceptors [42]. The general increases in δ and θ power are therefore indicative of a calming action of Adaptra^®^ Forte without inducing sedation.

Similar increases in δ and θ spectral power were observed in humans in the presence of theanine, a compound contained in tea [43]. Theanine is well known for its calming properties. Increases in spectral θ activity were reported for the synthetic drug buspirone [44]. Buspirone is prescribed for relieving anxiety and has been referred to as anxioselective. Thus, the observed general increase in θ activity in the presence of Adaptra^®^ Forte might be indicative of anxiolytic effectiveness. The focal increase in δ/θ spectral power during performance of a mental challenge indicates a positive effect of repetitive dosing of Adaptra^®^ Forte on concentration, since focal (opposite to general) enhancement of θ activity is related to changes in the attention system [45].

According to discriminant analysis, the efficacy of Adaptra^®^ Forte is projected not too far from and with a similar color to Lasea^®^, a herbal drug for relief of anxiety. Hence, Adaptra^®^ Forte might be used with a similar indication as Lasea^®^—namely, to treat anxious states. This result is in line with reports in the literature. One of the constituents of Adaptra^®^ Forte, *Withania somnifera*, is reported to be effective in the treatment of anxiety. According to [46], five clinical studies concluded that *W. somnifera* intervention resulted in greater score improvements (significantly, in most cases) than placebo in outcomes on anxiety or stress scales. Broad evidence from preclinical studies indicates a possible anxiolytic and/or anti-depressive activity of constituents from *W. somnifera* [26,47]. Regarding stress, some evidence in the literature indicates the capability of *Andrographis paniculata*, the second constituent of Adaptra^®^ Forte, to be effective in rats [48]. Thus, the combination of extracts from *W. somnifera* and *A. paniculata* seem to supplement each other, providing stress-coping and anxiolytic properties. Finally, a trend in improving cognitive functioning emerged during longer duration of treatment.

In conclusion, spectral changes in the quantitative EEG induced by Adaptra^®^ Forte seem to indicate calming as well as anxiolytic properties, possibly also resulting in patients better handling stress. Improvements with respect to mental challenges, despite the calming action, were only observed when comparing the difference between the first and last experimental day. The significance of the results could be improved in a future study by increasing the number of subjects. The results are in line with the adaptogenic properties that have previously been characterized for *Withania* and *Andrographis* preparations in the literature.

## 4. Materials and Methods

### 4.1. Participant Eligibility

The study was conducted in the Contact Research Organisation CRO NeuroCode AG (Wetzlar, Germany) with regulatory approval of an independent Ethical Committee at Landesärztekammer Hessen, im Vogelsgesang 3, D-60488 Frankfurt, Germany (Approval Date: 12 July, 2018; protocol number: PP_1717_EuroPharma_FINAL V2_15 May 2018; PP_1717_EuroPharma_FINAL V3_28 August 2018). The study was performed in accordance with the current version of the declaration of Helsinki (52nd WMA General Assembly, Edinburgh, Scotland, October 2000). The trial was conducted in agreement with the International Conference on Harmonization (ICH) guidelines on Good Clinical Practice (GCP). This trial is registered at ClinicalTrials.gov (Identifier: NCT03780621; https://www.clinicaltrials.gov/ct2/show/NCT03780621). All volunteers provided informed written consent to partake in the study. The investigator explained the aims, procedures, potential risks, anticipated benefits of the study, and provided a subject’s information sheet to each volunteer. The participants signed the consent form to state that the information had been explained and that they understood. They received a copy of the form and the original copy was maintained in a confidential file in the investigator’s records.

### 4.2. Selection of the Study Population

#### 4.2.1. Screening and Recruitment

Sixteen elderly volunteers of both genders aged 60 to 75 years were assessed for eligibility and enrolled in the study. Participants were eligible for participation in this trial in the period of December 2018 to May 2019. Patients taking dietary supplements that may affect cognitive functions were asked to terminate that treatment at least two weeks prior to beginning the study.

#### 4.2.2. Inclusion Criteria

The target population included elderly subjects experiencing mild cognitive impairment but who were otherwise healthy. The inclusion criteria were as follows:Male and female volunteers suffering from cognitive deficits.Questionnaire-DemTect. “DemTect” (for pre-selection of subjects) score values of 8–12 were regarded as conclusive.Age between 60 and 75 years inclusive.Subjects should be right-handed.Subject must be capable of providing informed consent.Acceptance of written consent to participate in the study after instruction in written and oral form (informed consent).

#### 4.2.3. Exclusion Criteria

Subjects were excluded having the following characteristics:Clinically relevant acute or chronic diseases as determined by case history or clinical examination.Clinically relevant allergic symptoms.Intake of clinically relevant medication in the last two weeks prior to screening (SC) as well as during the active study period based on notification by the subject or their case history.Intake of medication with primarily central nervous action (e.g., psychopharmaceuticals or centrally acting antihypertensives, antiepileptics, antidepressants).Known intolerance/hypersensitivity (allergy) to plant-derived extracts or any other of the ingredients of the investigational product (anamnestic survey).Presence of a rare genetic disease, such as fructose intolerance, glucose–galactose malabsorption, or sucrose–isomaltase deficiency (anamnestic).Intake of unusual quantities or abuse of coffee (more than 4 cups a day), tea (more than 4 cups a day), or tobacco (more than 20 cigarettes per day).Detection of alcohol at the time of the initial examination (day SC) or on study days A, B, C, D, and FU (positive alcohol test) or by case history.Smoking in the study center on study days A, B, C, D, and FU.Results of the DemTect Questionnaire score being <8 or >12.Participation in another clinical trial within the last 60 days.Poor compliance.Cancellation of informed consent.

#### 4.2.4. Withdrawal of Subjects

There were no withdrawals. The participants were free to leave the study at any time without explanations. They were advised that it would have no undesirable consequences.

### 4.3. Study Design

This was a randomized, double-blind, placebo-controlled, two-armed crossover trial of the fixed combination of *Andrographis paniculata* and *Withania somnifera* (Adaptra^®^ Forte, Europharma) compared with placebo on electric activity of brain (EAB) and cognitive functions of elderly subjects with cognitive deficits. The effects of Adaptra^®^ Forte and placebo capsules, taken for 4 weeks, were studied (Figure 7).

### 4.4. Intervention and Comparator

The dietary supplement used in this trial was a 550 mg Adaptra^®^ Forte capsule containing 400 mg of *A. paniculata* herb dry native extract (ratio of herbal substance to genuine herbal preparation 15–23:1, extraction solvent using 70% ethanol) standardized for the content of andrographolides (40 mg), and 150 mg of *Withania somnifera* roots and dry leaf extract Shoden^®^ (Arjuna Natural Pvt. Ltd, Kerala, India), (ratio of herbal substance to genuine herbal preparation 40:1, extraction solvent using 70% ethanol), standardized for the content of withanolide glycosides (50 mg).

The visually identical placebo capsules each contained 550 mg microcrystalline cellulose, corn starch, magnesium stearate, and brown sugar. The label included the drug name, study code, and storage conditions. Reference samples were retained and stored at Europharma USA (Green Bay, WI, USA).

#### Doses and Treatment Regimens

Participants received a package containing either Adaptra^®^ Forte capsules or placebo. They were instructed to take one capsule in the morning and one capsule at the evening with water for weeks. After four weeks of washout period, all participants started to take another test preparation for the next four-week period (Figure 7).

### 4.5. Allocation, Study Procedures, and Follow-Up

#### 4.5.1. Randomization, Blinding, and Allocation Concealment

Study preparations were randomly labeled by a qualified pharmacist (QP) and the random sequence of treatment codes was retained at the manufacturing site until the study was completed. The randomization code was kept secret from the investigators, and the code was only revealed after termination of the study. In this way, the investigators were blinded to the study medication and placebo control, thus ensuring a double-blind design. Study preparations were delivered to the study site pre-labeled and coded according to the random sequence. Participants were sequentially enrolled by the principal investigator (PI) at the study site. Participants’ allocation sequence (The Participants List), identifying the subjects and the study supplement packages (Treatment Code No.) were generated and maintained by the PI, who assigned each patient to a Treatment Code No. (from 1 to 16) and filled the patient name in the Case Report Form (CRF) and on the label of the package. The Participants List was used in statistical analysis at the end of the study together with the random sequence received from the QP. Information concerning the allocation of participants was kept in sequentially numbered and sealed envelopes that were stored by the chief executive officer (CEO) of the contract research organization.

#### 4.5.2. Evaluation of Compliance

The products under investigation were taken (days A, B, C, and D) under the supervision of the investigator and the time of application was documented (days A, B, C, and D). Therefore, compliance for all subjects on these experimental days was 100%. Each subject documented the medication intake daily with the date and the time of the application over the four weeks. From day A, day B (Phase A—4 weeks) and day C and day D (Phase B—4 weeks). The compliance was calculated by counting the remaining tablets for each subject. Overall, it was 99.53% (placebo = 99.72%; Adaptra^®^ Forte = 99.34%). The study monitor checked overall compliance with the study protocol upon their visits and the remaining capsules were counted at the end of the study. All unused capsules (two capsules in each package) were returned to the sponsor.

#### 4.5.3. Efficacy Measurement

At all visits, the patients were sitting alone in a quiet separate room with dimmed light in a comfortable chair. Baseline recording for 6 min under the condition of eyes open (Eo) was followed by 4 min eyes closed (Ec), 5 min d2-Test, 5 min ME-test, and 5 min CPT.

Test conditions were standardized and validated. EEG data were recorded twice: before (baseline) and 120 min (2 h) after ingesting the medication. Between the measurements, subjects spent their time in the facility’s recreation room. All experiments occurred at the same time each day (starting at 07:00).

##### Phase A, Screening and Training

On visit 1, the screening visit, participants were informed about the details of the pending study. All volunteers were checked for eligibility and signed written informed consent, including obligations that they would not take medicine or dietary supplements that may have potential effects on cognitive functions and would not consume more than one cup of coffee daily (in the morning) during the course of the study. After providing written informed consent and passing routine medical examination, eligible patients were included in the trial.

##### Phase A, Treatment and Assessment

On visit 2, participants passed all tests for cognitive functions and stress in the morning before intake of the trial medication (baseline). Then, the PI randomly assigned participants to the treatments and gave them the test compound. Participants orally took the Phase I treatment and repeated all tests 2 h after intake of trial medication. The treatment lasted 28 consecutive days. Participants took two capsules per day (one in the morning and one in the evening). Phase A treatment was followed by a washout period of 28 days.

##### Phase B, Treatment and Assessment

At visit 3, participants passed all tests for cognitive functions and stress in the morning before intake of trial medication (baseline). Then, participants orally took Phase II treatment and repeated all tests 2 h after intake of trial medication. Treatment lasted 28 consecutive days. Participants took two capsules per day (1 in the morning and 1 in the evening). Phase II treatment was followed by a washout period of 2 weeks. The study was then considered completed.

### 4.6. Efficacy and Safety Evaluation

#### 4.6.1. Efficacy Primary Outcome

The efficacy primary outcome measures were responses of electric brain activity as spectral power in 17 different brain regions within six specially defined frequency ranges (δ, θ, α1, α2, β1, and β2) during the test of cognitive performance: d2-Test, ME-Test, and CPT with financial reward. The EEG was recorded bipolarly from 17 surface electrodes according to the International 10–20 system with Cz as physical reference electrode (computer-aided topographical electroencephalometry: CATEEM^®^, MEWICON CATEEM-Tec GmbH, D94089 Altreichenau, Germany) using an electro-cap. For a detailed description of the procedure, please refer to references.

The signals of all 99 electrode positions (17 real and 82 virtual) were subject to fast Fourier transformation (FFT) based on 4 s sweeps of data epochs (Hanning window). Data were analyzed from 1.25 to 35 Hz using CATEEM^®^ software). In this software, the resulting frequency spectra are divided into six frequency bands: δ (1.25–4.50 Hz), θ (4.75–6.75 Hz), α1 (7.00–9.50 Hz), α2 (9.75–12.50 Hz), β1 (12.75–18.50 Hz), and β2 (18.75–35.00 Hz) [49]. This frequency analysis is based on absolute spectral power values and calculated as source density [50,51,52,53].

#### 4.6.2. Efficacy Secondary Outcomes

The efficacy secondary outcome measures were:The accuracy of processing the test for attention, concentration, and visual scanning speed, which is expressed as an error rate (ER) score (E%), is defined as the ratio of errors made to the total number of correct responses during 4 min and 40 s“Accuracy score” obtained in ME-TestConcentration performance score in CPTProfile of Mood States (POMS) using a validated POMS questionnaire [54,55,56]. The questionnaire contains 65 words/statements that describe people’s feelings. The test requires the patient to indicate for each word or statement how they have been feeling in the past week including today. Scores are: S1 = dejection (Niedergeschlagenheit); S2 = sullenness (Missmut); S3 = fatigue (Müdigkeit), S4 = thirst for action (Tatendrang).Quality of sleep score obtained by SF-B/R (Schlaffragebogen) questionnaire (a subjective assessment of sleep quality), used for quantitative and qualitative description and evaluation of sleep behavior and sleep experience. The SF-B/R comprises 31 questions and refers to the last two weeks [57,58,59]. The following subscores were used for the evaluation of the sleep questionnaire SF-B/R: SQ = sleep quality (Schlafqualität); GES = feeling refreshed after sleep/feeling of being recovered after sleep (Gefühl des Erholtseins nach dem Schlaf); PSYA = psychic well-balanced feeling in the evening (psychische Ausgeglichenheit vor dem Schlafenlegen); PSYE = psychic exhausted feeling in the evening (psychisches Erschöpftsein vor dem Schlafenlegen); PSS = psychosomatic problems during sleep (psychosomatische Symptome in der Schlafphase); TRME = dream recall (Traumerinnerung); SWR = sleep–wake regulation (Schlaf-Wach-Regulation).

### 4.7. Safety Outcomes

Safety outcome measures included incidence and severity of adverse events (AEs). The subjects were informed to report any AE occurring during the study to the PI or the PI’s personnel. Open-ended standardized AE questions were asked by the PI (or designee) during each contact with the subject. AEs observed or reported by a subject and/or staff member was recorded in the CRF. The following variables were recorded for each AE: description of the event, onset (date and time), resolution (date and time), maximum intensity, action taken, outcome, causality (yes or no), and whether or not it constitutes an serious AE. Safety outcome measures were assessed from the date of randomization until the end of the treatment.

### 4.8. Sample Size Considerations

We enrolled 16 patients (16 per treatment condition). Sample size was not determined during this pilot study. Estimation of the number of subjects to be included into the study was performed by considering data from earlier experimental results obtained under a similar experimental design.

### 4.9. Statistical Analysis

EEG data from the first recording session before intake of the capsules are reported as absolute numbers (µV^2^). For mathematical comparison to the results obtained for other preparations or drugs under comparable conditions, the linear discriminant analysis according to Fischer was used. Results from the first three discriminant functions were projected into space (x, y, and z coordinates), whereas results from the fourth to sixth discriminant functions were coded into red, green, and blue, respectively, followed by an additive color mixture (RGB-mode). To statistically document the different electric reactions of the brain to various cognitive loads, data from each challenge were documented as absolute spectral power (µV^2^). Adaptra^®^ Forte capsules versus placebo were compared by evaluation of the baseline recording of the last day in comparison to the baseline values of the first day (effect of single and repetitive dosing). Data from the first recording (baseline) were set as 100% and electrophysiological changes produced by placebo or Adaptra^®^ Forte capsules are reported as percentage changes. Since this study was an exploratory study with a small number of participants and EEG data were not normally distributed, the non-parametric sign test was chosen for comparing between the effects of placebo and Adaptra^®^ Forte. Exploratory statistics provided *p*-values.

## 5. Conclusions

In conclusion, spectral changes in the quantitative EEG induced by Adaptra^®^ Forte seem to indicate calming as well as anxiolytic properties, possibly also inducing better coping with stress. Improvements with respect to mental challenges despite the calming action were only observed when the difference between the first and last experimental day were regarded. Higher significance of the results is expected from a future study with a larger number of participants. The results are in line with the adaptogenic properties previously characterized for *Withania* and *Andrographis* preparations in the literature.

## Figures and Tables

**Figure 1 pharmaceuticals-13-00045-f001:**
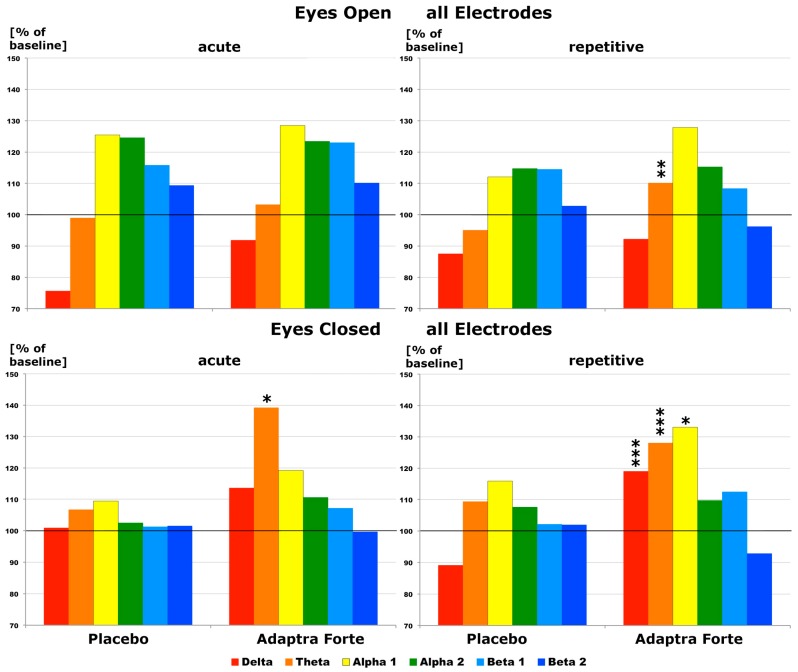
Effect of Adaptra^®^ Forte in comparison with placebo with regard to spectral changes in all brain regions and at all six frequency ranges (2 h after intake) regarding an average of all electrode positions during recording with either eyes open or eyes closed. Results from first experimental day (acute) on the left side, from the last day (repetitive) after 4 weeks of intake on the right side of the image. Baseline values were set as 100%. * *p* < 0.12; ** *p* < 0.05; ****p* < 0.01.

**Figure 2 pharmaceuticals-13-00045-f002:**
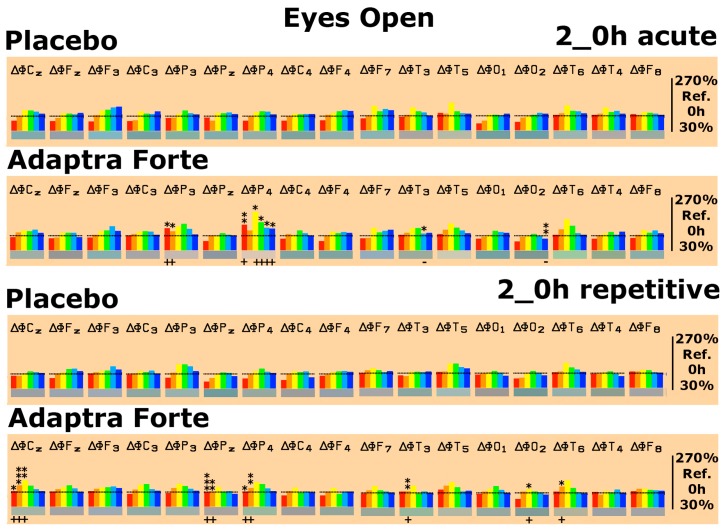
Spectral power differences under the eyes open recording condition on the first experimental day (acute) and on the last experimental day (repetitive) with administration of placebo or Adaptra^®^ Forte in all brain regions as represented by the electrode positions. C = central, *p* = parietal, O = occipital, F = frontal, T = temporal. Even numbers indicate location in the right hemisphere while odd numbers indicate the left hemisphere. Frequencies: red = δ, orange = θ, yellow = α1, green = α2, turquoise = β1, and blue = β2. The spectral power (between 30% and 270% on the ordinate of the bar graph) was averaged over 6 min and plotted against the pre-drug value (values at 0 min set as 100%), thus reflecting the effect of placebo or Adaptra^®^ Forte. * *p* < 0.12; ** *p* < 0.05; *** = *p* < 0.01 (non-parametric sign test) between placebo and Adaptra^®^ Forte. The direction of change is marked by + or − underneath the relevant bar.

**Figure 3 pharmaceuticals-13-00045-f003:**
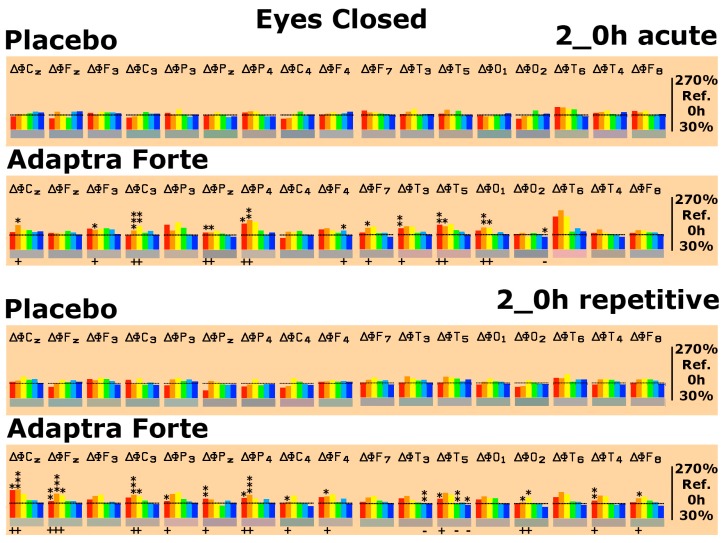
Spectral power differences under the eyes closed recording condition on the first experimental day (acute) and on the last experimental day (repetitive) with administration of placebo or Adaptra^®^ Forte in all brain regions as represented by the electrode positions. C = central, *p* = parietal, O = occipital, F = frontal, T = temporal. * *p* < 0.12; ** *p* < 0.05; *** *p* < 0.01.

**Figure 4 pharmaceuticals-13-00045-f004:**
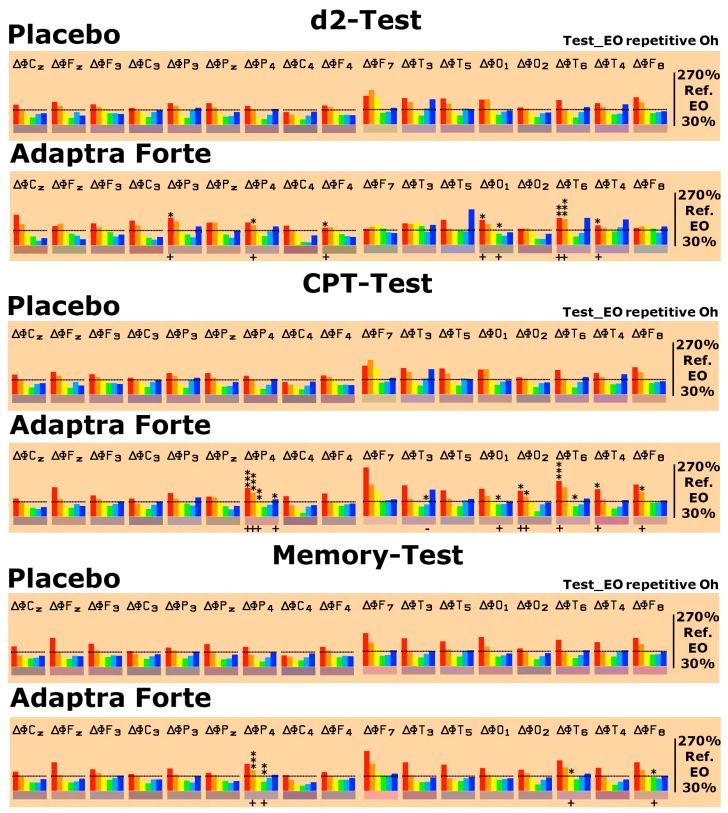
Spectral power differences on d2-attention-test, Concentration Performance Test CPT, and memory tests as deviation from the eyes open recording condition, which was set as 100% on the last experimental day with the administration of placebo or Adaptra^®^ Forte in all brain regions represented by the electrode positions. C = central, *p* = parietal, O = occipital, F = frontal, and T = temporal. Even numbers are located in the right hemisphere, uneven numbers in the left. Frequencies: red = δ, orange = θ, yellow = α1, green = α2, turquoise = β1, and blue = β2. The spectral power (between 30% and 270% on the ordinate of the bar graph) was averaged over 5 min and plotted against the eyes open condition (set as 100%), thus reflecting the effect of the placebo or Adaptra^®^ Forte. Statistical significance (non-parametric sign test) between placebo and Adaptra^®^ Forte: * *p* < 0.12; ** *p* < 0.05; *** *p* < 0.01.

**Figure 5 pharmaceuticals-13-00045-f005:**
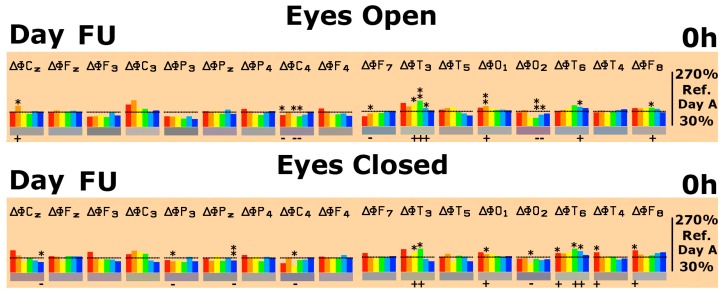
Data during eyes open and eyes closed conditions at baseline compared with follow-up (day FU) in comparison with day A (100%). Statistical significance between FU in comparison to day A is depicted for each brain region and each frequency range. Statistical difference in FU data is marked as: * *p* < 0.12; ** *p* < 0.05.

**Figure 6 pharmaceuticals-13-00045-f006:**
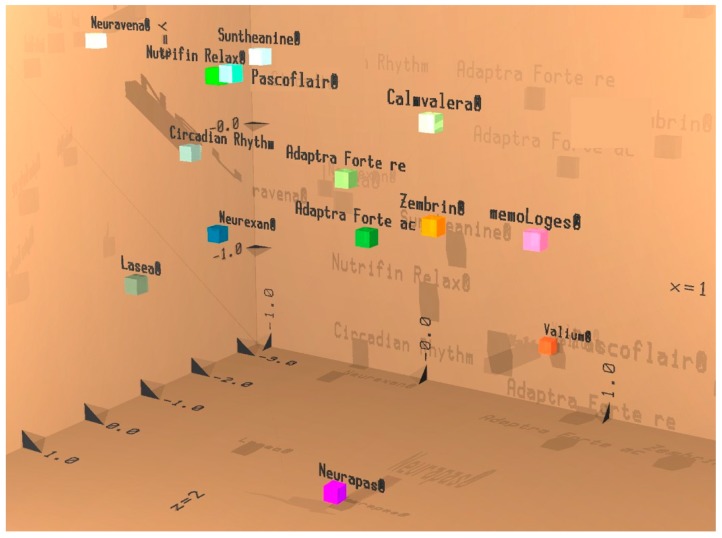
Result of discriminant analysis. Acute effect during eyes open recording condition: Adaptra Forte ac; repetitive effect during eyes open recording condition: Adaptra Forte re. Results from first 3 discriminant functions are displayed with respect to space (x, y, and z coordinates. Results from the next 3 discriminant functions are displayed as additive color mixture similarly to RGB mode on a TV. If preparations are projected in rather close proximity, they cannot well be discriminated from each other, which means they have a similar effect or can be used within a similar clinical indication.

**Figure 7 pharmaceuticals-13-00045-f007:**
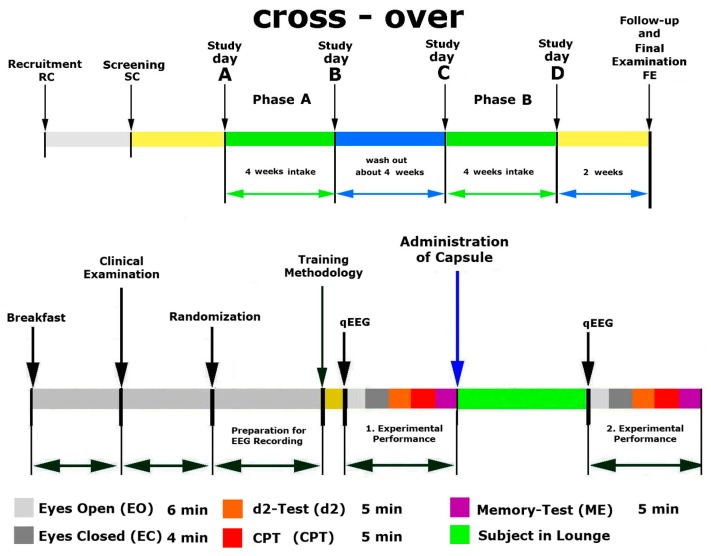
Design and timeline of the study days A, B, C, D, and FU. Performance: eyes open (Eo), eyes closed (Ec) and different cognitive tests: attention test (d2-Test), memory test (ME-Test), and concentration performance test (CPT).

**Table 1 pharmaceuticals-13-00045-t001:** Study population: baseline demographic characteristics.

Demographic Characteristics	*n*	Mean	SD
Age			
Total	16	63.81	2.04
Male	8	63.88	1.81
Female	8	63.75	2.38
Gender			
Total	16	11.00	1.11
Male	8	10.86	1.21
Female	8	11.14	1.07
Body height (cm)			
Total	16	173	0.10
Male	8	181	0.06
Female	8	164	0.06
Weight (kg)			
Total	16	78.03	14.92
Male	8	89.60	7.96
Female	8	66.50	10.46
Body Mass Index (BMI) (kg/cm^2^)			
Total	16	25.99	3.49
Male	8	27.33	2.60
Female	8	24.64	3.91

**Table 2 pharmaceuticals-13-00045-t002:** Absolute spectral power of baseline as recorded with eyes open. δ = delta, θ = theta, α1 = alpha1, α2 = alpha2, β1 = beta1, β2 = beta2 frequencies.

Absolute Values for “Eyes Open” at 0 h—First Day (Acute)												
δ			θ		α1		α2		β1		β2	
E	Placebo *n* = 15	Adaptra^®^ *n* = 15	Placebo *n* = 15	Adaptra^®^ *n* = 15	Placebo *n* = 15	Adaptra^®^ *n* = 15	Placebo *n* = 15	Adaptra^®^ *n* = 15	Placebo *n* = 15	Adaptra^®^ *n* = 15	Placebo *n* = 15	Adaptra^®^ *n* = 15
Cz	1.73	2.00	0.59	0.59	0.71	0.57	0.73	0.70	1.30	1.19	1.42	1.30
Fz	1.90	2.40	0.61	0.64	0.66	0.84	0.72	0.73	1.06	0.91	1.19	1.43
F3	2.05	3.09	0.79	0.78	1.00	1.18	1.04	0.98	2.38	2.15	4.29	4.39
C3	1.23	1.38	0.41	0.43	0.74	0.81	0.89	1.00	1.99	1.57	1.45	1.44
P3	1.01	1.78	0.35	0.49	0.76	0.96	1.02	1.15	1.08	1.13	0.84	0.85
Pz	0.99	1.17	0.35	0.33	0.51	0.53	0.64	0.62	0.80	0.69	0.55	0.55
P4	1.19	1.47	0.30	0.35	0.91	0.69	0.85	0.74	0.86	0.81	0.54	0.65
C4	1.18	1.46	0.38	0.38	0.72	0.82	1.08	1.31	2.07	2.37	1.78	2.22
F4	2.80	3.23	0.97	0.77	1.28	1.08	1.17	0.85	2.44	1.72	5.46	3.57
F7	7.23	10.74	1.33	1.76	3.38	2.59	2.82	2.02	3.26	3.68	4.23	6.45
T3	3.38	5.24	0.91	1.07	2.26	3.43	2.31	2.11	3.02	2.57	3.76	3.12
T5	2.75	3.46	1.08	1.23	3.17	3.00	2.90	2.30	2.53	2.95	2.07	1.99
O1	2.16	2.70	0.59	0.67	0.70	1.02	0.86	1.17	1.92	1.95	2.42	2.43
O2	2.07	2.57	0.61	0.58	0.83	0.87	1.14	1.28	2.39	2.56	2.48	1.97
T6	2.61	2.80	0.97	0.88	1.60	1.83	1.78	1.97	2.59	1.81	1.36	1.91
T4	2.66	3.38	0.85	0.87	1.97	2.45	2.11	2.77	3.98	3.68	3.77	4.90
F8	7.34	6.75	1.55	1.32	3.45	2.35	2.42	1.66	3.97	2.81	7.02	4.70
Med	2.16	2.40	0.79	0.67	0.91	1.19	1.02	1.28	1.68	1.72	2.16	2.34

**Table 3 pharmaceuticals-13-00045-t003:** Absolute spectral power of baseline as recorded with eyes closed. δ = delta, θ = theta, α1 = alpha1, α2 = alpha2, β1 = beta1, β2 = beta2 frequencies.

Absolute Values for “Eyes Closed” at 0 h—First Day (Acute)												
δ			θ		α1		α2		β1		β2	
E	Placebo n = 15	Adaptra^®^ n = 15	Placebo n = 15	Adaptra^®^ n = 15	Placebo n = 15	Adaptra^®^ n = 15	Placebo n = 15	Adaptra^®^ n = 15	Placebo n = 15	Adaptra^®^ n = 15	Placebo n = 15	Adaptra^®^ n = 15
**Cz**	2.00	1.96	0.71	0.73	0.81	0.98	0.99	0.94	1.81	1.50	1.39	1.55
**Fz**	2.25	2.45	0.72	0.87	0.96	1.25	0.84	1.09	0.95	1.09	0.90	0.97
**F3**	2.57	3.30	0.78	0.92	1.57	1.64	1.13	1.26	1.24	1.56	1.56	1.52
**C3**	1.23	1.58	0.50	0.52	0.79	1.21	1.12	1.35	1.70	1.92	1.37	1.45
**P3**	1.28	1.80	0.40	0.51	1.12	1.83	1.38	1.95	1.27	1.15	0.78	0.80
**Pz**	1.04	1.40	0.35	0.50	1.04	1.47	1.12	1.00	0.77	0.97	0.59	0.54
**P4**	1.13	1.64	0.38	0.47	1.05	1.39	0.97	1.51	0.85	0.77	0.48	0.58
**C4**	1.29	1.49	0.42	0.49	0.89	1.07	1.43	1.88	2.46	2.49	1.41	1.46
**F4**	2.76	2.87	0.97	0.87	1.44	1.72	1.08	1.30	1.48	1.47	1.56	1.30
**F7**	9.44	7.78	1.52	1.63	3.54	3.78	2.45	2.50	1.86	2.39	2.08	2.22
**T3**	3.19	3.56	1.00	1.14	2.38	2.99	2.27	2.25	1.98	2.54	1.67	2.00
**T5**	2.86	3.32	1.41	1.67	5.40	7.11	2.85	4.59	2.82	2.74	1.77	1.40
**O1**	2.49	2.19	0.77	0.74	1.06	1.92	1.29	1.81	2.08	1.89	1.55	1.54
**O2**	2.01	2.99	0.65	0.68	1.55	1.79	1.73	2.28	2.06	2.66	1.52	1.50
**T6**	2.77	3.40	1.31	1.57	5.87	10.60	3.81	3.51	2.28	2.62	1.67	1.43
**T4**	3.10	3.90	0.95	1.06	2.70	3.01	2.42	2.08	1.98	2.36	1.98	2.37
**F8**	8.07	6.00	1.83	1.46	3.72	3.41	2.98	2.67	2.07	2.22	2.38	2.23
**Med**	2.35	2.86	0.77	0.87	1.83	1.71	1.72	1.60	1.67	1.77	1.28	1.45

**Table 4 pharmaceuticals-13-00045-t004:** Change compared with the baseline in the psychometric scores (d2-Test, ME-Test, and CPT), profile of mood state rate (POMS), and the quality of sleep scores at the end of four weeks of treatment with Adaptra^®^ Forte compared with the placebo. Note that *n* = 14 for the d2-attention-Test at 0 h for Adaptra^®^ Forte.

Performance of Psychometric Tests									
	d2 Test			CPT			Memory Test		
	0 h	2 h	2 h vs. Baseline Two-tailed *t*-Test *p* <	0 h	2 h	2 h vs. Baseline Two-tailed*t*-Test *p* <	0h	2h	2 h vs. Baseline Two-tailed *t*-Test *p* <
	*n* = 15	*n* = 15		*n* = 15	*n* = 15		*n* = 15	*n* = 15	
Placebo repetitive–acute	0.48 (0.65)	0.39 (1.10)	0.799	−0.82 (2.45)	0.88 (2.45)	0.076	−1.14 (3.39)	−0.62 (4.68)	0.729
Adaptra^®^ Forte repetitive–acute	1.31 (1.16)	1.48 (1.44)	0.719	0.74 (3.73)	1.28 (3.68)	0.693	2.20 (3.24)	1.19 (4.28)	0.481
Placebo vs. Adaptra^®^ Forte two-tailed *t*-test *p* <	0.029	0.027		0.194	0.726		0.012	0.278	
Follow-up	12.01 (2.59)			4.49 (3.21)			9.34 (3.49)		
Values are means (SD)									

**Table 5 pharmaceuticals-13-00045-t005:** Results from (SF-B/R) sleep questionnaire. The questionnaire was filled out before the intake of the study medication. SQ, sleep quality; GES, feeling refreshed after sleep/feeling of being recovered after sleep; PSYA, psychic well-balanced feeling in the evening; PSYE, psychic exhausted feeling in the evening; PSS, psychosomatic symptoms during sleep; TRME, dream recall; SWR, sleep–wake regulation. Mean values are given ± standard deviation.

Evaluation of the SF-B/R Questionnaire (Schlaffragebogen)							
	SQ	GES	PSYA	PSYE	PSS	TRME	SWR
Placebo repetitive–acute (*n* = 15)	0.24 (0.35)	0.11 (0.24)	0.03 (0.41)	−0.18 (0.56)	−0.10 (0.44)	−0.13 (0.35)	0.40 (0.82)
Adaptra^®^ Forte repetitive–acute (*n* = 15)	−0.22 (0.72)	−0.25 (0.53)	−0.12 (0.77)	0.22 (0.69)	0.23 (0.31)	0.10 (0.57)	−0.05 (0.81)
Placebo vs. Adaptra^®^ Forte two-tailed *t*-test *p* <	0.039	0.028	0.510	0.094	0.024	0.192	0.145

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
