# Peer review of "Effects of an Adaptogenic Extract on Electrical Activity of the Brain in Elderly Subjects with Mild Cognitive Impairment: A Randomized, Double-Blind, Placebo-Controlled, Two-Armed Cross-Over Study"

_pharmaceuticals, 2020, doi:10.3390/ph13030045_

Round 1

Reviewer 1 Report

In subjected paper, the Authors presented studies upon the determination whether a combination of adaptogenic plant extracts from Andrographis paniculata and Withania somnifera (Adaptra® Forte) could be used as effective and safe treatment for impaired cognitive, memory, or learning ability functions and sleep disorders.

Authors presented this issue in a clear way by describing it correctly in a chapter “Materials and methods”. They also presented and discussed those results in chapters “Results” and “Discussion”.

Minor note:

  1. Adaptra® Forte is not a drug but a dietary supplement, so it should be clearly marked at work so that it doesn't mislead the reader. This should be clearly emphasized in the description of this plant preparation. In addition, the term dietary supplement should be used in the work text (methodology, results, discussion).

I would also like to add that the advisor is not a native speaker  and because of that I suggest that before having it published, language expert should read it.

Author Response

Ms. Ref. No.: pharmaceuticals-744946

Dear Reviewer

We would like to thank you for thoroughly reviewing the manuscript and their thoughtful criticism. We appreciate your suggestions and have revised the manuscript to address the comments and followed all suggestions to strengthen the manuscript.

Comment 1

The changes are marked by red color text as requested by the editor and submitted in separate files Adaptra® Forte is not a drug but a dietary supplement, so it should be clearly marked at work so that it doesn't mislead the reader. This should be clearly emphasized in the description of this plant preparation. In addition, the term dietary supplement should be used in the work text (methodology, results, discussion).

Response 1

In methods, page 16

The dietary supplement used in this trial was a 550 mg Adaptra® Forte capsule containing 400 mg of A. paniculata herb dry native extract (Drug Extraction Relation about 15–23:1, using 70% ethanol as extraction solvent) standardized for the content of andrographolides, 40 and 150 mg of Withania somnifera roots and dry leaf extract Shoden® (DER 40:1 (Arjuna Natural Pvt. Ltd, Kerala, India), using 70% ethanol as extraction solvent, standardized for the content of withanolide glycosides, 50 mg).

in introduction, page  2

The present study was initiated to test a combination of Andrographis paniculata and Withania somnifera (the dietary supplement Adaptra® Forte) in elderly subjects suffering from mild cognitive impairment. We hoped that the two preparations would interact positively with each other and provide adaptogenic efficacy within a clinical setting.

In results, Page 13

Overall, the treatment was well tolerated; no serious adverse events were detected in the presence of the dietary supplement Adaptra® Forte capsules.

Comment 2

I would also like to add that the advisor is not a native speaker and because of that I suggest that before having it published, language expert should read it.

This manuscript has undergone English language editing by MDPI before submitting. The text has been checked for correct use of grammar and common
technical terms, and edited to a level suitable for reporting research in a scholarly journal. MDPI uses experienced, native English speaking editors.

The certificate is attached

We hope that the revisions adequately address the concerns that were raised, and that the manuscript is acceptable for publication in special issue of Pharmaceuticals.

Best regards,

Alexander Panossian

Reviewer 2 Report

This manuscript highlights the use of adaptogenic plant extracts in electrical activity in the brain.  The authors used these extracts and found that they provide a calming effect without sedation.  These findings are interesting and provide a potentially meaningful use in the future.  

Author Response

Ms. Ref. No.: pharmaceuticals-744946

Dear Reviewer

We would like to thank you for thoroughly reviewing the manuscript. We appreciate your suggestions.

This manuscript has undergone English language editing by MDPI. The text has been checked for correct use of grammar and common technical terms, and edited to a level suitable for reporting research in a scholarly journal.
MDPI uses experienced, native English speaking editors. The certificate is in that attachment.

We hope that the revisions adequately address the concerns that were raised, and that the manuscript is acceptable for publication in special issue of Pharmaceuticals.

Best regards,

Alexander Panossian
